# Rethinking the Ideology of Using Digital Games to Increase Individual Interest in STEM

Shahrul Affendi Ishak [1], Rosseni Din [1,*], Nabilah Othman [1], Serge Gabarre [2] and Umi Azmah Hasran [3]

1   STEM Enculturation Research Centre, Faculty of Education, Universiti Kebangsaan Malaysia, Bangi 43600, Selangor, Malaysia; shahrulaffendiishak91@gmail.com (S.A.I.); cik.nabilah001@gmail.com (N.O.)
2   Department of Foreign Languages, College of Arts and Sciences, University of Nizwa, Nizwa PC 616, Oman; sergegabarre@unizwa.edu.om
3   Fuel Cell Institute, Universiti Kebangsaan Malaysia, Bangi 43600, Selangor, Malaysia; umi.h@ukm.edu.my
*   Correspondence: rosseni@ukm.edu.my; Tel.: +60-166-656-420

**Abstract:** Using games to teach science, technology, engineering, and mathematics (STEM) can enhance the quality of education. The idea of using digital games to increase individual interest towards STEM has been implemented as gaming technology has evolved in the 21st century. A correlation exists between using digital games and the learning outcomes which suggests that incorporating digital games may develop interest; however, the theoretical discussion on how interest may be developed with digital games remains inconclusive, thus, resulting in the need for further discussion. Hence, we aim to contribute with a discussion on how STEM digital games can develop individual interest in STEM. Previous studies on digital games in the STEM education context support the arguments presented in this study, therefore, there is a high probability that STEM digital games develop interest. Nevertheless, this can only be achieved with a good STEM game design and defining what is a good STEM game design is subjective. Several elements can be used as indicators to describe the quality of a design. These include the pedagogical learning content and the inclusion of fun elements within a design. Therefore, we propose the integration of theories with pedagogy, learning strategies, STEM learning content, game elements, and game principles design to ensure the presence of a three-layer process to develop interest. The creativity of game designers and developers is key to creating appealing STEM digital games providing young players with an inspiring experience.

**Keywords:** STEM digital games; STEM interest; play; game design; 21st century pedagogy

## 1. Introduction

Digital gaming is one of the strategies for better quality education towards fulfilling the 2030 goal of sustainable development. Since the early 2000s and with the technological developments in graphical systems, digital games have been identified as one of the best pedagogical approaches for education. This has led researchers and educators to incorporate such games into the educational context. Educators from the field of STEM (science, technology, engineering, and mathematics) pioneered their use to enhance the teaching and learning in their classrooms. This field of study remains dynamic and current as witnessed by the emergence of several theories laying the framework to develop impactful STEM educational digital games.

Within the goal of sustainable development, education plays a big role in developing humankind's knowledge that may lead to future world technologies and civilizations. STEM education consists of the core fundamental subjects needed to achieve this goal. Over the past 20 years, as technology was incorporated into STEM education, digital games caught the attention of educational technologists. Thus far, studies indicate that digital games are beneficial to the cognitive development, particularly in the learning process. As digital games are part of the modern entertainment culture, they have a high propensity to be utilized. Integrating suitable STEM learning contents into digital games can help sustain

and enculture STEM learning in meaningful and playful environments. This pedagogical strategy can enhance the application of existing STEM knowledge among children after classroom hours, surpassing the games' entertainment purposes.

Playing games improves one's cognitive ability and both educational and non-educational games have an impactful experience if they are successfully designed and developed. Neuroscientists have demonstrated that digital games have a significant effect on brain functions [1,2] and it has been shown that such games positively impact physical, cognitive, social, and emotional development. By harnessing their huge potential for STEM education, game developers and educational technologists seek to provide gaming environments that can target these human developmental aspects, particularly the cognitive development that is associated with STEM learning.

Digital games are developed for specific purposes. These purposes include learning, training, medical therapy and entertainment and game design takes into account a high universality and usability to ensure that a large number of people can benefit. Identifying what makes a good game is highly subjective; however, aesthetic design, usability and the playability of a game are key to getting people's attention and delivering a better gaming experience. There is no exact formula to design good educational digital games due to the nature of creative products. Moreover, each player or user may interpret a game differently [3]. Consequently, most digital games rely on guidelines for their conception and design. Scholars mix input elements and games into the design to achieve a specific output and studies reveal that these mix inputs include pedagogical elements for learning, simulations for training skills, psychological interventions for medical therapy, and fun for entertainment purposes. Scholars have also identified thousands of results and models in this field of study; however, very few if any discuss how to design digital games to increase interest towards STEM.

Most studies on digital games conducted in the STEM education context report positive empirical results in terms of learning improvement [4–12]. Two decades of rapid technological development on digital gaming show that using digital games is one of the best pedagogical approaches to enhance STEM teaching and learning. The current digital games evolved from being console-based towards mobile-based platforms over the past decades. This shows that exposure and the popularity of digital games are beneficial to 21st century learners; however, STEM educators do not solely emphasize learning achievements, as students should also be able to connect the knowledge they gain with real-world practice.

There is an issue as the positive results presented in most studies on STEM-based digital games, including their development, usability and implication, are not reflected in the current school students' perception of STEM. Moreover, STEM education with science and mathematics are perceived as the hardest and most unpopular subjects in school [13,14]. The level of interest towards STEM subjects remains low in most countries. This is also the case in Malaysia where the government and the news report the decreasing number of STEM major enrolments in upper secondary schools and in higher education [15]. We believe that such a situation may be avoided as digital games can influence STEM students' performance and stimulate their interest.

We gathered information from the literature on digital games and noted that there are three main areas of focus in digital game studies. These include (1) their development [16–20], (2) their design and usability [21–23], and (3) their implication for the human developmental state [24–27]. Only a few recent articles [28,29] deal with STEM digital games or STEM educational digital games. Educational digital games are usually associated with users in the K-12 educational [30,31] system And, commonly, STEM education is confined to secondary schools. There is a huge knowledge gap centered on what STEM digital games are and why most authors suggest that digital games have the potential to increase interest towards STEM.

Most authors suggest that digital games have the potential to trigger STEM interest. We argue that something may be wrong with the games' design which reduces their effectiveness. Since design is a crucial part of digital games, a good design should deliver

a better gaming experience for all users. Thus, we suggest that the ideology of using digital games be revised and rethought to solve the issue of a lack of interest towards STEM. The discussion on how digital games should act as a mechanism to develop STEM individual interest is presented in this article. Several relevant topics dealing with the nature of play, of digital games as appealing objects to spark interest, of how STEM digital games should be designed, and the future direction for implementation ensuring quality education are presented.

## 2. Nature of Play

Playing is associated with children as it has a major impact on their development. The peak of playing occurs at the middle childhood stage and decreases as children get older [32]. Children explore and learn through games. Piaget and Vygotsky are the two most influential theorists who emphasize the role of playing in children's development. Both argue that playing provides more opportunities for children to interact with materials in their environment. This interaction through play allows them to construct their own knowledge of the world and with the advent of modern technology, playing has shifted to digital platforms. This shift is also due to tightly structured families, school schedules, parents working outside of homes, and the lack of safe spaces to play. Currently, digital games represent one of the main sources of playful activity and entertainment.

In the 21st century, digital games are considered a popular form of activity for children as well as adults. Their ease of access and mobility allow people to play on their smartphones. Although playing such games might be viewed as individualist, studies indicate that online digital games allow individuals to connect with others in virtual online worlds. Moreover, neuroscientists report that playing is associated with stress reduction [2,33]. As can be seen in Table 1, most people play games for fun, challenges, storyline/narratives, to compete online, and to escape from reality [34]. In the educational context, digital games provide a place and time to learn that is not achieved by completing a school worksheet.

**Table 1.** Reasons why people play digital games and the type of game platforms used. Adapted from GWI [34].

| Reason to Play | Game Console (%) | Handheld Gaming Device (%) | PC/ Laptop (%) | Smartphone (%) | VR Headset (%) |
|---|---|---|---|---|---|
| For fun | 70 | 60 | 67 | 65 | 61 |
| For the challenge | 36 | 38 | 34 | 31 | 48 |
| For the storyline/narrative | 28 | 33 | 22 | 18 | 36 |
| To compete online | 23 | 28 | 22 | 19 | 41 |
| To escape from reality | 30 | 30 | 26 | 23 | 37 |

Nowadays, people spend their leisure time playing digital games. The highest number of digital game users is found in the urban population of big cities, particularly among adults. The number of smartphones constantly increases resulting in a total of 2.5 billion mobile gamers across the world [35]. In 2017, most digital gamers were 21–35 years old (28% men and 19% women). Children also play digital games on a daily basis [36]. Findings from studies among children conducted in the United Kingdom between 2013 and 2018 reveal that children aged 3–15 years old spend a daily average of 6.2–13.8 h playing digital games [37]. The Statista Research Department reports that every year the number of hours played increases slightly.

Playing helps develop psychological, cognitive, physical, social, emotional and literacy skills. Nowadays, it is common for individuals, especially children, to prefer playing digital games over reading books. Studies indicate that playing digital games creates entertaining and fun experiences and the reasons for games' popularity have been studied from a scientific perspective. People report that digital games make them feel happy and entertained, as confirmed by studies in neuroscience which indicate that dopamine levels

are higher when playing digital games [33]. Playing digital games has a positive effect on each part of the brain which includes learning improvement [38], health condition, better eye coordination and movement [39], memory ability [40], motivation boost [41] and the speed of attention shift [42].

## 3. Digital Games as Appealing Objects

People around the world enjoy playing digital games, especially children. Advanced graphics and interactive systems in the digital games make them more attractive, engaging, appealing and entertaining. Studies have been conducted to identify the reasons why children find playing digital games so entertaining, with most findings suggesting that playing digital games allows players to experience virtual worlds where they are empowered beyond their usual control.

### 3.1. High Engagement

Digital games provide engaging gaming experiences. The self-determination theory in digital games explains why people love to play digital games. Edward Deci and Richard Ryan first introduced this idea in 1985 explaining human intrinsic motivation and behavior [43]. Scholars in the field of digital games view this theory as having a major contribution in explaining this phenomenon. Digital games are engaging because they address autonomy (the ability to make decisions), competence (the ability to do something successfully or efficiently), and relatedness (the state of being related or connected). Reflecting on digital games, characters and virtual worlds make games more appealing. They provide players with platforms where they can be part of these worlds, where players have the opportunity to take on new roles and become their ideal selves. These new roles and identities could have a different gender, be a hero or a villain and this results in improved feelings about oneself.

Animation allows any form of art to appear as moving images and provides players with the freedom to control characters or elements within a digital game. The idea of gaming experiences from the educational technology perspective was introduced in 2002 by Garris [44]. When playing games, players interact through the game mechanics with their judgment, behavior and system feedback and this continuous game cycle occurs until the game is finished. Interactive worlds within digital games create gaming experiences. These virtual worlds are defined as a "magic circle" [45], and provide players with their own set of rules to reach the games' goals.

### 3.2. Visual Appeal

The graphics and the visual designs of digital games contribute more than experiences towards the appeal factor since seeing something different beyond reality is more engaging and appealing. Malone's [46] idea on intrinsic motivation is viewed as a major contributing factor towards game elements. He argues that the appeal of computer games is linked to challenge, curiosity, and fantasy. Players tend to explore and try to overcome all the challenges set in a game' and curiosity increases concomitantly with the games' levels. Moreover, elements of fantasy make the game more attractive as people can interact with objects beyond their imagination and these ideas are developed through game elements including goals, rules, narrative, feedback, challenge, interaction, and characters.

### 3.3. Debriefing Knowledge

Integrating learning content into game mechanics provides not only a gaming experience but also leads to individual knowledge construction by enabling players to indirectly construct their understanding. Game designers want meaning, facts, and information to be stored in a player's memory as the games provide more than a fun and entertaining experience. Garris, Ahlers, and Driskell [44] propose the idea of integrating learning content with game elements to create games specifically tailored to achieve learning outcomes. In the STEM context, integrating STEM-related topics or concepts allows players to experience

and gain a declarative knowledge of these topics. We strongly believe that a good gaming experience and an interactive visual design might explain why STEM digital games can stimulate the players' interest in STEM.

Knowledge gained by players is a causal relationship construct of using STEM digital games. Playing digital games stimulates each part of the brain to trigger individual thinking, planning, problem solving, emotions, decision making, and behavioral control [33] leading to cognitive development and learning achievements. As reported by several studies, improvement on the STEM conceptual understanding is the main knowledge gained from such games [10,47]; however, two types of learning achievements are identified in knowledge gain. These are declarative knowledge (knowing what) and procedural knowledge (knowing how) [48]. This level of knowledge relates to the brain's development stage. For adults, information received while playing games can be used in the games' worlds but are also significantly related to real-world applications; however, this might not be the case with children. In the early stages of development, children might perceive information within the game as what it is, meaning that the procedural knowledge might be subliminal. Children grab scientific facts and concepts from specific STEM-related topics within the game world, while understanding their purpose for real world applications. They might not experience it in the present time, but this conceptual understanding remains valuable [29] and this can lead to a sustained interest in STEM.

### 3.4. Mechanisms Influencing Interest

STEM digital games are the context or object for the players. Players decide whether a game is appealing or interesting from their first attempt and interest is a psychological construct. Therefore, it may present good or bad impressions towards a digital game. Social background, intrinsic motivation, past experiences, and self-determination are factors influencing individual perceptions of digital games; however, this interest can be harnessed with a good STEM design. Hence, game designers need to create engaging interactive STEM gaming experiences to stimulate player interest in STEM through engaging interactions and gameplay. There is a strong interdisciplinarity between the various STEM subjects, thus, a suitable interactive game design should allow children to experience this interdisciplinarity to solve real-world problems.

Individual interest can be developed when using digital games. Krapp has explained individual interest development which is expanded in the present article [49]. Interest development is first triggered by the environment when a person interacts with objects they find appealing. The interaction then forms a situational interest where the person starts to engage with these objects over time. Then, this situational interest is later internalized and developed into individual interest. Krapp defines this notion as an ontogenetic transition [49]. This notion is furthered as individual interest development is a process which undergoes four phases of development: phase 1, triggered situational interest; phase 2, maintained situational interest; phase 3, emerging individual interest; and phase 4, well-developed individual interest [50,51]. Findings from a recent study suggest that digital games can be used to stimulate interest in STEM with primary school children [29]. We propose that children's interactions with STEM digital games develop individual interest if they feel connected and engaged with the games (Figure 1).

The success of using STEM digital games to stimulate interest in STEM is conditional on the gaming experiences and interactive visual designs while different players may have different judgments and perceptions. These cause the development of interest to cease at the beginning of a game if a game does not have a sufficient impact to stimulate their interest in STEM (Figure 2). Players are exposed to STEM digital games on their first attempt when the game begins. They have their first impression of the game based on the visuals, mechanics, gameplay system, characters, and playability. If a STEM digital game does not impress them, they stop playing it; however, in other cases players may try further, to explore and complete the game. Engagement of a player with a game determines the successful attribute of STEM interest development. Hence, game designers and developers

should pay attention to factors that may hinder that interest development. Abiding to design requirements helps avoid such issues and produces good STEM digital games that can stimulate an interest in STEM among children.

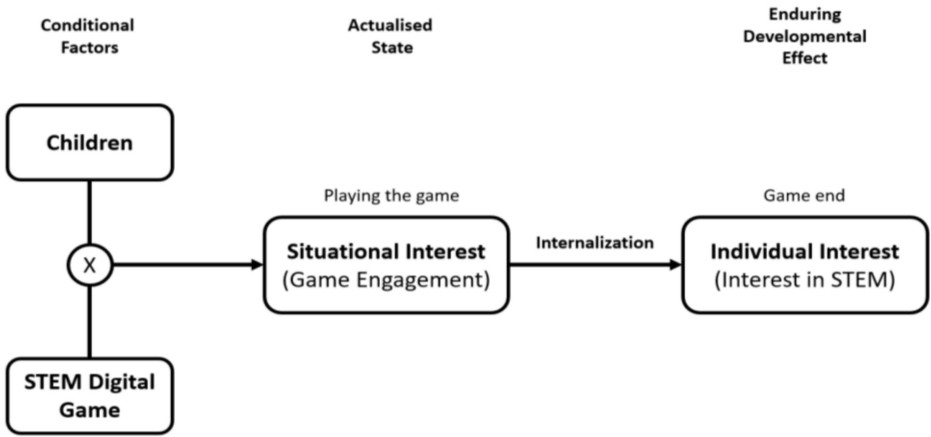

**Figure 1.** The ontogenetic transition from situational to individual interest via STEM digital games [29].

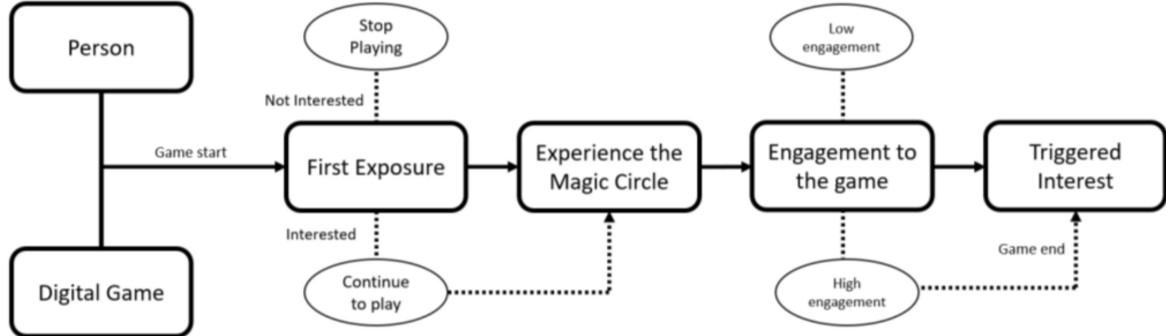

**Figure 2.** Proposed path flow of a player's interaction within STEM digital games.

## 4. Towards Good STEM Digital Games

Good games depend on how well they are designed and developed. By looking at the nature of digital games and their possible use to stimulate the development of interest, we suggest that game designers use these core concepts and incorporate them into the design of digital games. STEM digital games should deliver integrated STEM educational content while providing a meaningful gaming experience which triggers interest. Studies indicate that meaningful experiences are the main draw to playing digital games [52–54]. Meaningful gaming experiences happen when players are highly engaged in a game's world [28,29] with which they have a strong and deep connection. Hence, game designers and developers should strive to engage players. To achieve this, good designs for STEM digital games should be emphasized.

What defines a good digital game is subject to interpretation. It constitutes an art form developed by creative production teams. Different game designers incorporate different concepts and ideas in their designs. Moreover, a game's success is subject to its end users' perceived playability, interest, and acceptability. A game is ready for the commercial market when it reaches its target.

Studies on game design emerged with the early developments of the first digital games. They increased in popularity as the potential use of digital games for educational purposes emerged, particularly in STEM education. Digital games are produced every day with creativity and rigor. This situation also occurs in academia when researchers integrate STEM learning content into digital game mechanics through gamification. Research in this field furthers the knowledge on how to design a good educational digital game.

### 4.1. Designing a Digital Game

Since the advent of digital games, designers have tried to create intrinsically motivating gaming experiences [55]. The key to developing a game is to ensure that it is both fun and entertaining. A growing number of game studies categorize game elements into essential elements and game systems. Essential elements of digital games include players (single or multiplayer), information, actions, payoff (rewards), objectives (goals) and rules. These five elements are the core requirements for a game. Game system is the secondary essential element that focuses on the appearance of each digital game. It is derived from the creative expression of designers and programmers who create better gaming experiences. Game systems are categorized as conflict, strategy, aesthetics, story (narrative), fantasy, feedback, and challenges.

To create a good gaming experience, players should feel a sense of accomplishment [56,57]. This results in higher motivation as the players continue playing. This well-known concept presented by Csikszentmihalyi [58] and known as the flow theory is presented in Figure 3. Designers should identify a clear set of goals, a good balance of challenges for players skills and clear immediate feedback. Furthermore, a good game should keep players in the state of flow. The challenge of a game against a player's set of skills should be directly proportional. As such, challenges should not be too difficult when a player's skills are low as they may cause anxiety; however, challenges that are too easy may result in player boredom.

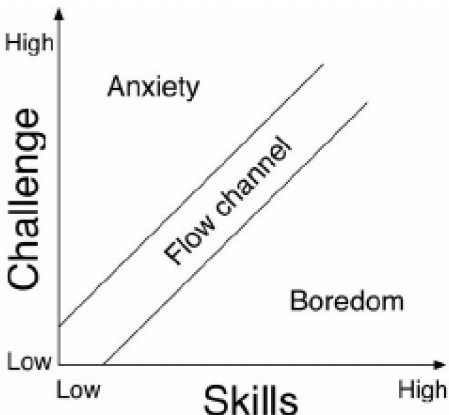

**Figure 3.** The balance of challenges against the players skill as proposed in the psychological theory of flow.

There is no single right way to design a game [3]. Instead of an ideal blueprint, the fundamentals of good game designs are mainly referred to as guidelines and most guidelines for designing digital games have an emphasis on game elements. The early mechanics-dynamics-aesthetics (MDA) framework has been used in most games' designs and development. The MDA framework helps guide designers to create playable game mechanics and to provide fun to end users [59]. Gradually, learning content became the core part of educational digital games, while in educational technology studies, learning content is part of the design elements. Several models are proposed based on the different developmental objectives of games, such as education, therapy, or entertainment. Most recent studies tend to propose their own framework and guidelines for good game designs [11,23,55,57,60–69]; however, none of these studies focus on the design of STEM digital games. These should be designed in a way that develops interest in STEM.

By conducting a review of the literature, we found that the development of digital games follows two perspectives: the game design (entertainment-based oriented) and the educational technology design (education and entertainment-based oriented). Specialists in the field of game design have advanced design and programming skills and they use fundamental guidelines to produce games. Whereas educational technologists study how games affect users' learning achievements. They emphasize the input requirements to

create effective digital games for learning. Consequently, these two perspectives should be combined to provide a holistic framework that benefits those disciplines [28,29]. In the STEM digital games context, there are few input models from game elements fundamentals, universal designs for instructional products, or game-based learning. Hence, in summary the input and its principle indicator for a good STEM digital game should consist of theories, pedagogy, learning strategies, STEM learning content, game elements and game design principles (Table 2).

**Table 2.** Summarization of input specification requirements for STEM digital games.

| Input | Principle Indicator | Description | References |
|---|---|---|---|
| Theories * | Experiential learning theory | Players experience the four stages of the learning process while playing games. These consist of concrete experience, reflective observation, abstract conceptualization and active experimentation. | [1,28,29,70,71] |
| | Educational-psychological theory of interest development | While playing, interaction between players provides appeal and engagement in the gaming world to develop interest through four phases which are triggered situational interest, maintained situational interest, emerging individual interest, and well-develop individual interest. | [28,29,49–51] |
| | Self-determination theory | Engagement of players with the gaming world is caused by strong autonomy, competence, and relatedness of a designed digital game. | [28,29,53,72] |
| Pedagogy ** | Problem-based learning | Each game has a problem that needs to be solved by players, in the form of missions. The problem is designed through several mechanics. Integrating problems in a game is important to help determine the learning outcomes. | [73–76] |
| Learning Strategies ** | Self-directed learning | Players use their set of skills and develop understanding towards the designed gaming world. The players' intention is to solve the problems designed into the digital games. They use skills such as observing, testing, and drawing conclusions. This form of self-directed learning is continuously used until the end of the game (problem solved). | [6,57,75,76] |
| STEM Learning Content ** | Any STEM related topic | Any STEM topic can be integrated into game mechanics. The specific learning content needs first to be identified and to undergo a gamification process. This process ensures that the content is correctly transformed into playable game mechanics. | [5,28,30,77] |
| Game Elements ** | Players | The person who controls an entire game in the designed gaming world. | [57,78,79] |
| | Information | A set of information is designed, stored, and presented to the players in the gaming world. | [60,80,81] |
| | Actions | Actions taken based on the decisions made by the players. | [31,82] |
| | Payoff | Payoff is also known as rewards after players complete missions. It can exist at every level of the game and as a final reward, after missions are accomplished by the players. | [57,59] |
| | Objective (Goal) | Goals are a major part of games. Problems are designed by determining what goals players should achieve in a game. There is only one specific goal in a digital game. | [30,57] |

**Table 2.** *Cont.*

| Input | Principle Indicator | Description | References |
|---|---|---|---|
| | Rules | Players must obey a set of rules designed into a game. Players lose points and need to restart the game if they break any rules. | [57,83] |
| | Conflict | Conflicts are presented to the players as they interact with a game system. Usually, they are portrayed by non-player characters (NPC), and mostly villains. They try to stop the players from reaching the next level or achieving their goals. | [84] |
| | Strategy | Strategy is a set of decisions made by the players after they observe, test, and draw conclusions within a game. A wise planned strategy might help players pass all obstacles and win the game. | [85] |
| | Aesthetic | Desirable emotional responses evoked by players while interacting with the game system. It plays a major part of the gaming experience to ensure a game is appealing to the players. It prompts players to continue playing a game. A game's graphical style mostly refers to the aesthetic. | [59,86–88] |
| | Story | Stories in games represent the characters (avatars) and how the characters develop in the games. The experiences of the characters are controlled by the players. | [57,89–93] |
| | Fantasy | Fantasy is part of fun and engaging elements. Games allow the representation of any form of fantasy. Good, visualized characters and sceneries in a game enable players to feel excited about something unreal that they can see and virtually interact with. | [46,94] |
| | Feedback | Feedback is the monitoring progress of players towards the games' goals. It can be in the form of remaining life, energy, time, and location. Feedback allows players to receive more explanation on actions taken, especially if they are taking a wrong step and lose in a specific level. | [41,57,95] |
| | Challenges | Challenges are designed to test the players towards gaining their mastery. When players master all challenges, purpose no longer exists. Challenges become increasingly difficult as a game progresses. | [57,96,97] |
| Game Principle Design ** | Player's cognitive ability | Players have their own cognitive abilities. Unlike children, teenagers and adults can explore any type of games. Designers should pay attention to a game's design, so that it is not too difficult for children as they would lose interest from not being able to solve the problems. | [30,60,96,98] |
| | Gender inclusiveness | Gender inclusiveness gives a higher possibility for games to be played. Each game design should minimize gender stereotypes in resolving conflicts, entertainment criteria, and responses. | [99–101] |
| | Layout design | The layout design should be suited to the players' hand gestures and eye coordination. | [102] |
| | Value | The game should embody good value that avoids any inappropriate violence and sexual content. | [60,103] |

Note: * Important elements; ** Elements that need to be integrated with the applied theory.

*4.2. Integrating STEM Learning Content*

As can be seen in Figure 4, there are thousands of STEM-based games applications which can be downloaded from Apple's App Store and Google Play, by using a STEM game keywords search. Most games are categorized as educational digital games; however,

there are a few popular commercial games (non-educational digital games) that are associated with STEM related concepts. Recent commercial digital games applications have better graphic designs and playability but require longer production times. Unlike game applications, most educational games are designed by researchers or by small, local game companies. Among these are *Mystemville*, *Hero Elementary*, *Learn to Code*, *Beats Empire*, *MarcoPolo Weather*, *RoboCo*, *Slice Fractions*, and *mPower Math*, but educational digital games are less favored over casual and commercial games. Studies suggest that some non-educational digital games such as *Plant* vs. *Zombie*, *Angry Birds*, *SimCity*, *Foldit* and *Minecraft* may enhance STEM skills [10,12,45,104]. Table 3 presents examples of STEM-based digital games with STEM learning outcomes.

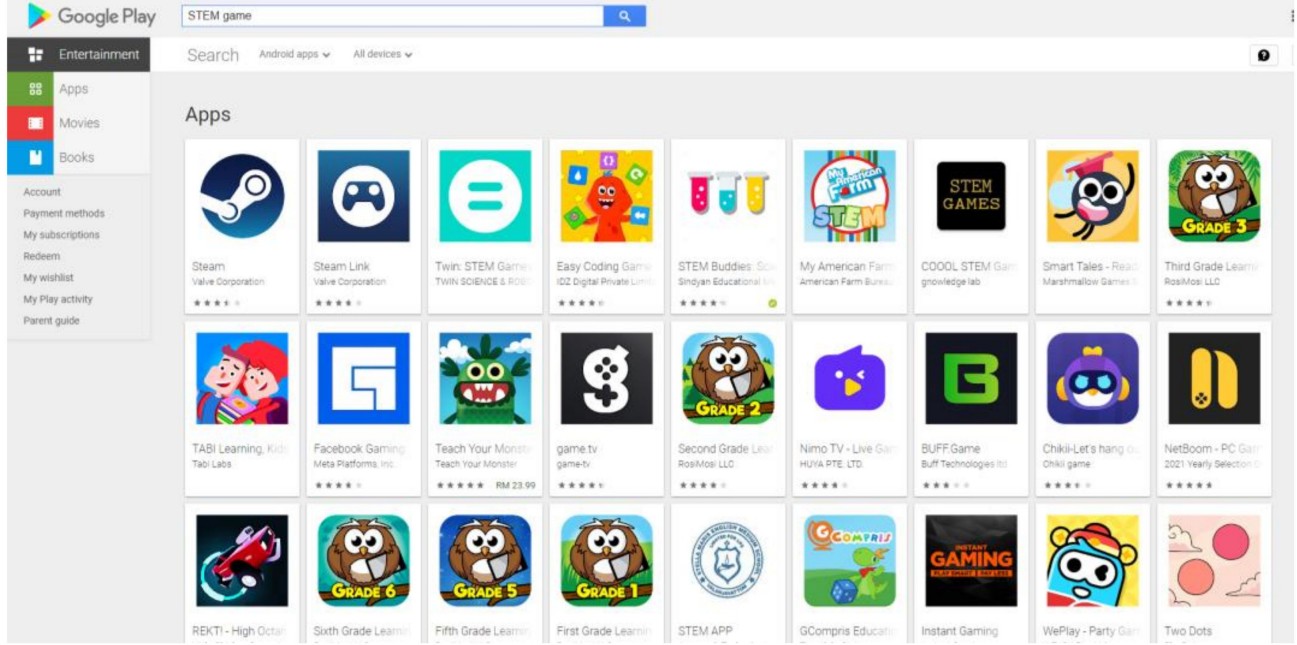

**Figure 4.** Search result for "STEM game" in the *Google Play Store*.

When designing and developing digital games, STEM related topics are ordinarily integrated as individual disciplines. Science and mathematics are more common than engineering and technology. Indeed, digital games with a focus on technology are rare. Most studies are conducted on the importance of technology to design and program games requiring coding. This is reflected in the design of games for the two core STEM syllabi of science and mathematics, whereas engineering and technology are usually combined into one concept. Most importantly, designers should integrate correlated STEM disciplines without losing each discipline's unique depth, characteristics, and rigor [105].

The idea of integrating STEM related contents for digital games is a good starting point to expand the theoretical foundation underlying good design and development. The lack of valuable STEM digital games, as demonstrated by most studies, emphasizes the absence of interdisciplinary STEM elements. The interrelation of STEM elements should be highlighted as part of the overall design, so that players can understand the usefulness of applying knowledge from each STEM discipline to drive the entire game; however, the integration of STEM content into game mechanics (objective and goal, feedback, challenges, levels, progress bar, rewards and badges, timer or countdown, competition, leaderboard) allows players to interact with a set of rules and feedback loops. This interaction with the system produces an enjoyable STEM gameplay. To implement all interdisciplinary STEM disciplines in a game, Leung (2019) suggests emphasizing the central role of science and mathematics. Engineering acts as a requirement for researching, designing, and producing, whereas the application of robotics, coding and programming are technology oriented.

**Table 3.** Example of STEM-based digital games with STEM learning outcomes.

| STEM-Based Digital Games | Outcome |
| --- | --- |
| *Angry Birds* | This game uses a catapult concept to teach the physics of positioning to hit a target. |
| *Plant* vs. *Zombie* | This game train players on strategic planning and mathematical skills by using the arrangement of plants against zombies. |
| *SimCity* | This game allows players to use their architectural and urban planning skills. |
| *Minecraft* | This popular game among children trains their engineering and architectural skills to design block units and form space. |
| *Foldit* | This online citizen-science game allows players to deal with protein molecules. Scores are based on the structure of proteins that are successfully folded. |
| *Mystemville* | This game allows players to train their scientific knowledge and skills on farm management. |
| *MarcoPolo Weather* | This game trains players on the meteorological concepts by controlling nine weather conditions. |
| *Hero Elementary* | This game challenges players to deal with real-world problems by applying their STEM knowledge. |
| *Learn to Code* | Players learn how to use coding and game developmental concepts by designing characters, background, animation, and coding. |
| *Beats Empire* | Players enhance their computational thinking skills by running a music studio and record label in a fictional city reminiscent of New York. |
| *Slice Fractions* | This game enhances the knowledge of fractions in mathematics. |
| *RoboCo* | This game allows players to use their engineering skills to design a robot to serve the needs of a future world. |
| *mPower Math* | This game enhances the players' positive attitude and confidence and helps to develop the foundation of STEM skills. |

### 4.3. Designing STEM Digital Games

Based on the input elements from Table 2, we propose an integrated model considering both digital game and educational technology design perspectives purposely to increase interest towards STEM. This integrated model stems from the observation that some games are designed with an emphasis on game elements as input to deliver a better gaming experience over learning theories, while others prioritize learning over the game elements. This could be the result of game design teams not having enough expertise in game design or when the gaming industry ventures into educational games without considering the STEM educational contents. We believe that this issue can be resolved with a precise universal model of STEM game design with an emphasis on both perspectives of digital game design (game elements and game principle design) and educational technology (learning theory, pedagogy, learning strategy, values, and STEM learning content).

This model is mainly adapted from the UDin universal design, the agile development model, and a combination of selected principles of game design found in the relevant literature. The UDin model is the current model for developing any type of digital based teaching tool [28,29,76,106]. Common non-educational digital games lack educational content as their purpose is to be fun and entertaining; however, the integration of STEM-related educational content into the game design will transform games from being non-educational to educational. The integrated learning content does not stand alone. Other pedagogical elements such as learning theories, values, pedagogy and learning strategies are required to ensure that the mechanics of the designed STEM content are suited to the players' cognitive abilities, but most importantly, learning output is achieved. STEM digital games are not only created to deliver learning content but also to provide a STEM gaming experience that will stimulate interest in the subject.

STEM digital games should provide a meaningful gaming experience to increase the players' interest towards STEM. Among the six game design elements proposed in this article, the integration of theory is crucial in shaping STEM digital games. Theory determines

how mechanics and gameplay lead to the development objectives. The core constructs of this pedagogical approach are STEM digital games, STEM learning, meaningful gaming experiences and STEM interest [28]. The three most relevant theories to game design stimulating STEM interest are the experiential learning theory, the educational-psychological theory of interest development [49–51], and the self-determination theory [46]. To develop interest through digital games, a three-layer process should occur when playing a game. As can be seen in Figure 5, integrating design elements ensures the presence of this three-layer process.

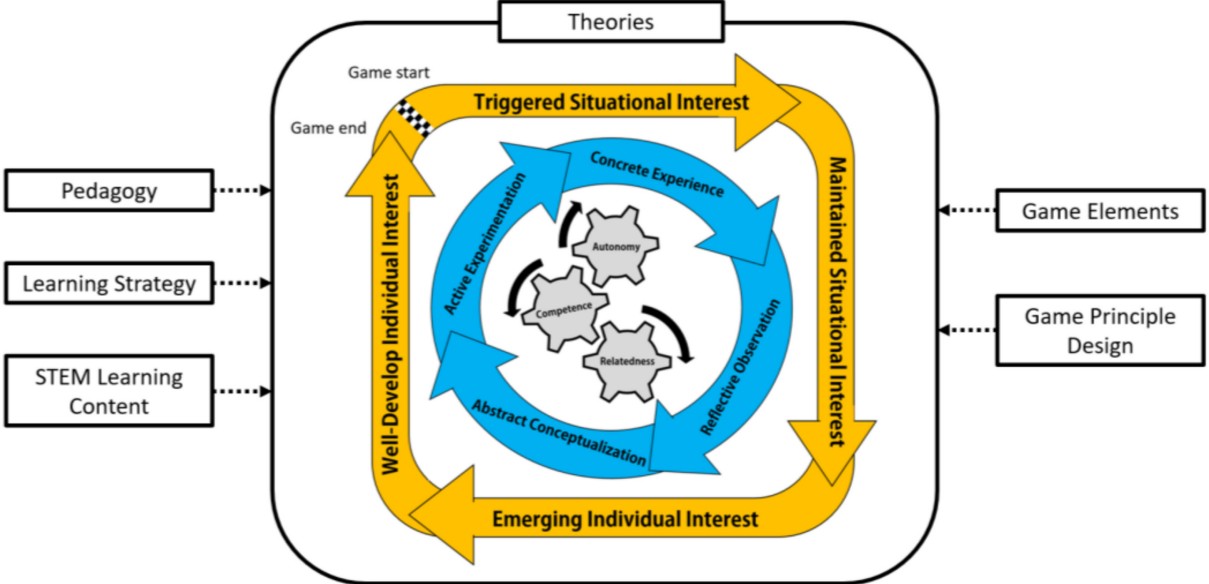

**Figure 5.** Three-layer process to stimulate STEM interest with STEM digital games.

STEM digital games provide an environment that supports the individual pursuit of interest towards STEM. As such, interest towards STEM can be developed by playing STEM digital games. Integrating theories (experiential learning theory, educational-psychological theory of interest development, and self-determination theory) with pedagogy, learning strategy, STEM learning content, game elements, and game principle design forms a set of gameplays within STEM digital games. Players interacting with a game are in a 'magic cycle' and continue to experience it until the game ends [45]. Experience and interaction only occur while in the game. Players need to make judgements to determine their actions and move through decisions. Accordingly, the consequences of their actions in the game contributes to their understanding of the game's mechanics to achieve the main goal. Being in the 'magic cycle' while interacting with gamified STEM learning contents through game mechanics helps form a conceptual understanding, commonly represented as learning outcomes.

Learning outcomes are individually constructed by players through their interactions while in the 'magic cycle' [29,44]. The construction of knowledge and understanding of STEM learning contents occur in four phases: (1) concrete experience (players first interact with a game, (2) reflective observation (players observe the game's system for the first time where inconsistencies between understanding and experience are noteworthy), (3) abstract conceptualization (players generate new ideas pertaining to the modification of abstract concepts of STEM topics), and (4) active experimentation (players apply this idea to the game world to see the results). The four learning phases of the learning cycle happen continuously, especially when a new level is introduced to a player.

Interacting with a game system alone does not trigger the learning experience, as this is also the result of self-determination. Players continue to experience a game because of their sense of autonomy, competence, and relatedness with a STEM digital game. A good game design empowers players to explore and shape a game's narrative (autonomy),

stimulates them to reach mastery through progression (competence), and allows players to feel connected with characters and the game's story.

In Figure 5, the two inner processes guarantee that learning and a meaningful gaming experience occur while players are in the gaming world. The outer layer is the subliminal result of learning and engagement. Consequently, the heart of a good game is dependent on the ability to keep players willing to continue playing. Without this, the three-layer process does not occur. Game designers and developers should ensure that their games have a high level of usability and appeal to increase interest through stimulation, while meaningful and engaging STEM games can trigger situational interest.

Situational interest can evolve to individual interest through game flow as players are introduced to increasing levels of challenges and new dimensions as the game progresses. Game duration and engagement are essential to the emergence of individual interest. Players gather information and construct their own understanding from the gamified STEM learning content. This results in the awareness of the scientific concepts that are designed in a game. Thus, the knowledge gained is useful to proceed successfully to the end of the game. As a result, experience and being part of a gaming world while playing STEM digital games should inspire players in their lives.

In the last stage, emerging individual interest becomes well-developed. Players feel a strong engagement and empowerment as characters in the game through a developmental phase. Their connection to the STEM digital games (object of interest) becomes more stable and generalizable. Hence, players understand the use of STEM for real-world problems (as presented in the games) and the interest in STEM is well-developed. Players make conscious choices and may autonomously pursue their interest in STEM related activities. Moreover, by playing games, users may decide to opt for STEM career paths.

## 5. The Way Forward and Future Perspectives

As edutainment digital media, STEM digital games should provide meaningful gaming experiences to allow knowledge construction that will result in interest in STEM. Early stages of design and development are important to conceptualize the specific input needed. Providing a platform that people, and especially children, can relate to contributes to achieving educational goals. Even though interest towards STEM is the focus of early-stage STEM education, continuously providing STEM experiences with digital games may develop and sustain interest. Most importantly, by engaging in games, players become eager to learn more about specific STEM subjects. Consequently, they are more involved in STEM-related activities which can lead them towards a career in STEM.

With this article, we aim to set the foundations for how STEM digital games play a role in stimulating interest in STEM. We advocate that design and development are the basis to achieve this goal. Good game designs that integrate theories, learning strategies, pedagogy, STEM learning content, values, game elements, and game principle design as universal attributes are essential to delivering good STEM digital games. Furthermore, we strive to shift the use of digital games to outside schools. Indeed, the school setting is not practical for a digital game-based pedagogy approach due to the lack of digital devices and to the already heavy syllabi. Children already learn STEM concepts in schools, and digital games can enhance their knowledge through fun interactive visuals (game world) with concealed scientific notions and applications.

Instead of focusing on the use of digital games to ensure quality education, we should shift the way we look at digital games which have a potential for individual cognitive development. A proper game design and STEM learning content integration can provide a good platform to inspire people with STEM. Consequently, game designers should strive to produce more quality STEM digital games. They should be aware of the nature of play and how digital games should appeal to players.

By doing so, game designers should come out with creative ideas and adopt the elements of good game design as theoretically presented in this article and children are the best target to implement this pedagogical approach. Their higher level of curiosity while

interacting with STEM digital games provides a suitable opportunity to develop STEM interest and to sustain it. To achieve this goal, game designers and developers should work alongside STEM teachers and educational technologists in a multidisciplinary approach. Therefore, the three-layer process proposed in this article should be a starting point to empirically explore digital games in a STEM context.

## 6. Conclusions

The concept of using digital games as a pedagogical approach to increase interest in STEM needs to be revised. In this article, we argue that digital games can positively influence people. To achieve this, digital games should consist of proper gamified STEM learning contents that will define what STEM digital games are. With good designs, STEM digital games can be appealing to players, and especially to children. The integration of theories with pedagogy, learning strategies, STEM learning content, game elements, and game principle design ensures interest and its development. In this article, we propose that interest towards STEM can be developed with the simultaneous presence of learning experiences and the self-determination of players while in a game. We propose six STEM game design elements that serve the three-layer process of developing STEM interest with digital games. This novel idea of digital games in the STEM education context needs to be further explored and studied empirically.

**Author Contributions:** For this research article, all author contributions to this study specifically were: conceptualization, S.A.I. and R.D.; writing—original draft preparation, S.A.I.; review and editing, S.A.I., R.D., S.G. and N.O.; supervision, R.D. and U.A.H.; project administration and formatting, N.O. All authors have read and agreed to the published version of the manuscript.

**Funding:** We would like to thank the STEM Enculturation Research Center and Universiti Kebangsaan Malaysia for supporting this research and publication with grant code GG-2021-014.

**Institutional Review Board Statement:** Not applicable.

**Informed Consent Statement:** Not applicable.

**Data Availability Statement:** Data sharing is not applicable.

**Acknowledgments:** We would like to convey our utmost appreciation to the STEM Enculturation Research Centre, Faculty of Education, Universiti Kebangsaan Malaysia and all researchers who have contributed in various ways.

**Conflicts of Interest:** The authors declare no conflict of interest.

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
