# Peer review of "Rethinking the Ideology of Using Digital Games to Increase Individual Interest in STEM"

_sustainability, doi:10.3390/su14084519_

Round 1

Reviewer 1 Report

Dear Authors,

The topic of your article is very interesting. I agree with you regarding the paucity of Digital Games supporting STEM education and expect to find some good examples of Digital games aimed to teach or at least motivate children to learn STEM. Unfortunately, there are no such examples. It is essential to incorporate in the article some good examples. 

You mentioned: "Educational digital games are less preferable game to be played as compared to casual and commercial game. Some studies suggest non-educational digital games also suitable to enhance STEM skills like Plant vs. Zombie, Angry Birds, SimCity, Foldit and Minecraft." Unfortunately, there is no citation and no explanation of how these games can contribute to STEM studies.

I see the main contribution of the article in that it presents in an organized way the game's requirements (Table 2) and tries to connect the game models to learning theories. However, I did not see it as a great innovation compared to the many sources that deal with the subject. The only difference concerns a recommendation to introduce STEM content in games. 

Your article gives some theoretical framework regarding Digital Games requirements and guidelines for developers. However, the main issue is implementing the STEM content in these games, and this question is not answered.

There are many general statements without proper references, diminishing the overall impression of the article's academic level. 

The article begins with general claims that have no basis in sources. For example, the statement "Most digital game studies conducted in STEM education context indicate positive empirical result in term of learning improvement" needs to be referenced. The statement "We gather all information from existing selected literature on digital games" sounds too pretentious. 

You mentioned, "Edward Deci and Richard Ryan who first introduced this idea in 1985 regarding human intrinsic motivation and 156 behaviour" with no citation.

The statement like "In STEM context, integrating STEM-related topic or concept allow player to experience and gain declarative knowledge towards the topic" is very promising, however, there is no explanation how it may be realized.

The statement "Several models were proposed based on the developmental objective of the game 306 either for education, therapy, or entertainment. Most recent studies tend to propose their 307 own framework and guidelines for good game design" also require citations.

Regarding the language – there are numerous grammatical mistakes. I recommend proofreading by a native English speaker.

Best regards!

Author Response

Dear Respected Reviewer,

We would like to inform that the correction has been made according to the reader's comment.  Attached is the table of correction and rebuttal.  

I had also done the proofreading before sending back the final corrected manuscript.  If it need to be proofread again I will send it to proofreading services suggested by the journal once the acceptance is received.

Thank you for your kind attention.  If you have further inquiries, I would be glad to answer them. 

Associate Professor Rosseni Din
Corresponding Author

Reviewer 2 Report

The paper is an interesting and worthwhile review of the potential use of digital games in STEM instruction. Yet, key issues exist regarding its format and content.

Format:

Widespread errors exist in the syntactic structure of sentences and word choices. Also, the logical order of sentences does not seem to fit the authors' intended meaning.

STEM is an acronym. It should be spelled out first. Then, the acronym can be used consistently.

Content:

In the abstract, the statement "good STEM game design" needs to be operationalized.  

In the introduction, the extent to which the use of digital games (as a pedagogical choice) is related to sustainable education needs to be clarified and expanded.  What are the specific properties of a well-designed game for STEM learning? Are there differences in its quality that are linked to the specific materials and skills to be learned?

In Line 254, "meaningful game experience" should be given an operational definition.

Overall, the purported benefits of digital games for STEM learning need to be clarified. Most importantly, a distinction needs to be made between interest in an activity and learning of a specific subject matter (including skills and knowledge).  Both are worthwhile goals, but one does not automatically lead to the other.

Are there limitations to the use of games in STEM learning?  In the end, the reader needs to be convinced that particular designs are effective tools for stimulating students' interest (i.e., cognitive, affective, and behavioral engagement), as well as fostering learning. The paper may need to be re-written to ensure that both aims are attained. The reader also needs to be convinced that the principles of a “good STEM game design” overcome the differences in the materials and skills to be learned in STEM disciplines.  

Author Response

(The authors gave the same response as above.)

Round 2

Reviewer 1 Report

Dear Authors,

The topic of your article is very interesting. I agree with you regarding the paucity of Digital Games supporting STEM education and expect to find some good examples of Digital games aimed to teach or at least motivate children to learn STEM. Unfortunately, there are no such examples. It is essential to incorporate in the article some good examples. 

You mentioned: "Educational digital games are less preferable game to be played as compared to casual and commercial game. Some studies suggest non-educational digital games also suitable to enhance STEM skills like Plant vs. Zombie, Angry Birds, SimCity, Foldit and Minecraft." Unfortunately, there is no citation and no explanation of how these games can contribute to STEM studies.

I see the main contribution of the article in that it presents in an organized way the game's requirements (Table 2) and tries to connect the game models to learning theories. However, I did not see it as a great innovation compared to the many sources that deal with the subject. The only difference concerns a recommendation to introduce STEM content in games. 

Your article gives some theoretical framework regarding Digital Games requirements and guidelines for developers. However, the main issue is implementing the STEM content in these games, and this question is not answered.

There are many general statements without proper references, diminishing the overall impression of the article's academic level. 

The article begins with general claims that have no basis in sources. For example, the statement "Most digital game studies conducted in STEM education context indicate positive empirical result in term of learning improvement" needs to be referenced. The statement "We gather all information from existing selected literature on digital games" sounds too pretentious. 

You mentioned, "Edward Deci and Richard Ryan who first introduced this idea in 1985 regarding human intrinsic motivation and 156 behaviour" with no citation.

The statement like "In STEM context, integrating STEM-related topic or concept allow player to experience and gain declarative knowledge towards the topic" is very promising, however, there is no explanation how it may be realized.

The statement "Several models were proposed based on the developmental objective of the game 306 either for education, therapy, or entertainment. Most recent studies tend to propose their 307 own framework and guidelines for good game design" also require citations.

Regarding the language – there are numerous grammatical mistakes. I recommend proofreading by a native English speaker.

Best regards!

Author Response

Dear Respected Reviewer,

We would like to inform that the correction has been made according to the reader's comment. Please see the attachment. Attached is the table of correction and rebuttal. The proofreading had also done by Serge before sending back the final corrected manuscript. 

Thank you for your kind attention. If you have further inquiries, I would be glad to answer them.

Associate Professor Rosseni Din
Corresponding Author

Reviewer 2 Report

Please proofread the entire document again ensuring proper syntax and word choices.

Author Response

Dear Respected Reviewer,

We would like to inform that the correction has been made according to the reader's comment. Attached is the table of correction and rebuttal. This paper has been undergone proofread by native English speaker.

Thank you for your kind attention. If you have further inquiries, I would be glad to answer them.

Reviewer’s Comment

Author’s Respond

Author

Please proofread the entire document again ensuring proper syntax and word choices.

This paper has been undergone proofread by native English speaker.

Dr Rosseni/ Nabilah

Associate Professor Rosseni Din
Corresponding Author

Round 3

Reviewer 1 Report

I think the manuscript is improved.